# How to Rate the Financial Performance of Private Companies? A Tailored Integrated Rating Methodology Applied to North-Eastern Italian Districts

Guido Max Mantovani [1,2,3,*] and Gregory Gadzinski [1]

1   Department of Economics and Finance, International University of Monaco, 98000 Monaco, Monaco
2   The Teofilo Intato Institute, 20129 Milan, Italy
3   Department of Management, Ca' Foscari University of Venice, 30123 Venice, Italy
*   Correspondence: gmantovani@monaco.edu

**Abstract:** This paper contributes to solving the puzzle of assessing the financial performance of private/unlisted companies. The inner characteristics of these companies make the adoption of traditional best practices in estimating risk premia difficult or impossible. Moreover, the lack of market data and comparable information biases the perception of corporate performance and generates the misallocation of credit fundings (both quantities and pricing). Hence, in this paper, we develop an Integrated Rating Methodology (IRM) to estimate a more efficient corporate "return-to-risk" measure. Our IRM is rooted in the seminal "certainty equivalent" model as developed by Lintner in 1965, but we modify it using a shortfall approach, and then compute a "confident equivalent" that is compliant with Fischer Black's zero-beta model as well as the Basel agreements. An empirical application of the approach is conducted with a sample of 13,583 non-financial SMEs in the north-east regions of Italy, where there is evidence of inefficient bank financing. We back-test our IRM by rating these companies using corporate financial data during the period 2007–2014, which encompasses both the Great Financial Crisis and the European sovereign debt crisis. Our empirical results depict a clear crowding-out effect of credit allocations when we compare our IRM scoring measure with the actual raising ability and the cost of capital relating to these firms. We find that 36% of companies are underfunded, even if they have a superior IRM score, while 27% of them are funded without merit. Interestingly, this last figure is in line with the average non-performing loan ratio provided by official Italian statistics from 2015 to 2020. Therefore, we conclude that our IRM methodology is promising and may be better at estimating risk financing in small private companies (including start-ups) than internal banking models. These initial results will drive our forthcoming research towards creating an IRM 2.0.

**Keywords:** SME financing; rating; certainty equivalent; Basel regulation

**JEL Classification:** G32; M10; G28

## 1. Introduction

Something is still wrong in current standards for scoring private/unlisted companies, particularly when they are in relation to Small and Medium Enterprises (SMEs). Accordingly, the need to adopt innovative methodologies is imperative. Biases in assessing the economic and financial performances of these firms are probably at the root of the misallocation of funds for both Equity and Debt financing. Standard financial practices are based on hypotheses that widely differ from the reality of these businesses, such as, for example: (i) investment assessment is based on the marginal contribution of the asset to an already well-diversified portfolio; (ii) the nature of risk is given (i.e., it is supposed to be exogenous) so that the asset sensitivity to non-diversifiable risk is not considered; and (iii) risk premia are computed through betas, frequently estimated using peer-group analysis.

The debt capital markets are no exceptions. The regulation on credit scoring and rating seemed (and still appears) unable to prevent the allocation of bank allowances to poorly performing companies. The main reason for this is the short-termism that backs up these methodologies: the focus on the (12-month) probability of default signals a preference for liquidity indicators, while less attention is given to the long-term performance of firms, and therefore also to the persistence and solvency of corporate returns (Nigro and Dennis 2005).

The main purpose of this paper is to apply the Integrated Rating Methodology (IRM) introduced in Mantovani (2014) as an innovative system for assessing corporate performance, i.e., rating/scoring small and medium firms while bypassing the above biases. The IRM focuses on the comprehensive corporate ability to maintain persistent returns in the long run, using an integrated approach that infers the financial attractiveness of the firms' business models. The IRM is based on Lintner's seminal work on certainty equivalents (Lintner 1965), from which we derive an original extension that eases its practical implementation and therefore avoids the problems that practitioners encounter when estimating the risk premia.

To obtain concrete evidence of the IRM efficacy and its true applicability, the methodology was back-tested on credit allowances provided to manufacturing and service firms in the North-East regions of Italy during the years 2007–2014. In fact, while Italian SMEs represent an international benchmark for their competitiveness and distinctive hallmarks, the North-Eastern ones have suffered from a lack of bank financing, although they are among the key forces of Italian economic growth[1]. Figure 1 demonstrates the abnormal dynamics of non-performing loans (NPLs) in north-eastern Italy, a region with one of the highest densities of private companies and SMEs. The significant jump in 2017 is proof of the above pitfalls.

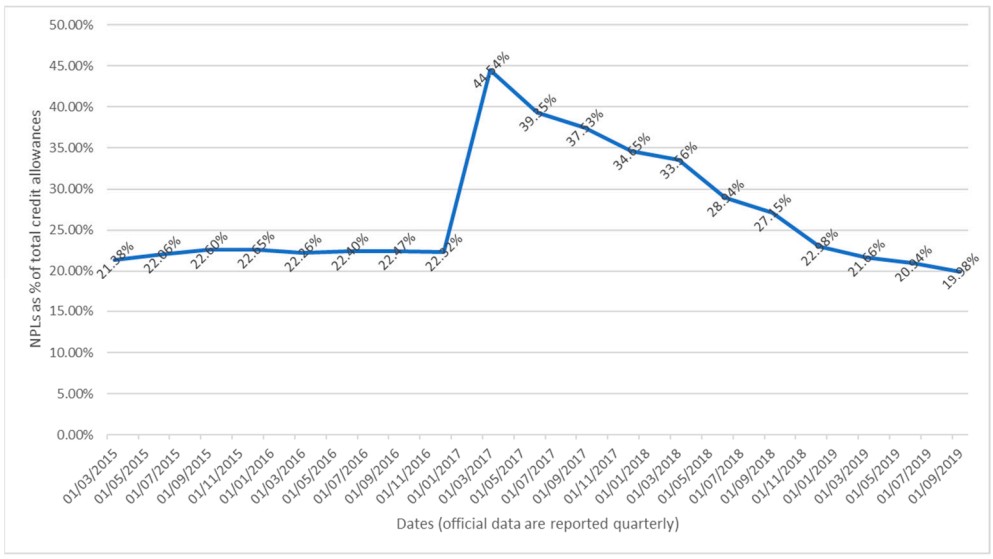

**Figure 1.** Dynamics of NPLs as % of total credit allowance in north-east Italy. The figures in the chart are the authors' computations on original data as sourced from the official public database of the Bank of Italy.

The empirical evidence in this paper demonstrates the magnitude of credit misallocation in Italian SMEs and Private Firms. According to our results, 36% are underfunded, based on their risk-adjusted performance, while 27% are funded, although without merit of credit. This empirical analysis also demonstrates that past corporate performances matter more than their perspectives in the actual allocation of bank allowances. In fact, no clear evidence of a relationship between firm performance (i.e., both profitability and leverage ratio) and overall corporate risks was found. This casts the regulating framework in an embarrassing light, since it contributes to the diversion of capital flows from sound investment opportunities. This is consistent with some literature (Masschelein 2003; Allen et al. 2004;

Berger et al. 2005) that advocates the use of efficient rating systems to improve credit channelling and then foster economic growth.

This paper is organized as follows: Section 2 presents a literature review on the efficiency of banking credit allowances. Section 3 illustrates the model, while Section 4 describes the actual use of the model in the Italian case by showing and discussing the empirical results. Section 5 concludes this study.

## 2. Basel Agreements and the Efficiency of Debt-Capital Markets: A Literature Review

The Basel Committee of Banking Supervision was founded in 1974 by the central bank governors of the Group of Ten (G10) countries. Its aim was to increase financial stability and the quality of banking supervision worldwide by setting minimum standards for the regulation and the supervision of banks. Capital adequacy became soon the inner focus of the committee's activities, and the Basel Capital Accord (also known as Basel I) was approved in 1988. It required banking institutions to have a minimum equity capital of the banking institution vs. its risk-weighted assets, with a ratio fixed at 8%. In 1999, the committee issued a new Revised Capital Accord (Basel II), which entered into force in 2004. The new framework was designed to better reflect the quality of underlying assets (i.e., their risks) and address financial innovations that had occurred in recent years. The changes aimed to reward and encourage continuous improvements in risk measurement and control[2]. After the dramatic global financial crisis, the Basel Committee released another accord in 2010, Basel III, with the purpose of further improving the resilience of financial markets[3]. The implementation of Basel III began in January 2014, although it was limited to the risk-based capital requirements. The concept behind the Basel agreements is very clear and well diffused in financial literacy: the greater the risk, the higher the capital at risk must be. Unfortunately, the nature of risk and its sources make risk measurement difficult for business purposes (Castellan and Mantovani 2016). In fact, risk measurement is a commonality in the updates to the agreements.

Since the introduction of Basel II, it has been predicted that (at least large) banks would adopt an internal-rating-based (IRB) system, thus substituting the standardized approach (SA). Based on an analysis of Belgian banks, Masschelein (2003) concludes that IRBs seem to imply lower capital requirements thanks to greater efficiency in measuring risks. Similarly, Allen et al. (2004) show that adopting a credit scoring system allows for a faster and less costly investment valuation. Berger et al. (2005), Cowan and Cowan (2006), and Frame et al. (2001) go a step further and, in analysing the US market, conclude that adopting an external credit scoring system increases SMEs' financing. For instance, Berger et al. (2005) found, in a sample of US banks, that the adoption of an external credit system contributes to a significant increase in SMEs' financing over a three-year cycle. Similar results have also been reached by Cowan and Cowan (2006), who used a survey methodology for their analysis, and by Frame et al. (2001).

An additional issue involves determining which methodology to implement for the rating system, i.e., quantitative, qualitative, or a mix of both. The literature related to quantitative analysis is relatively well developed and mainly concerns models for corporate bankruptcy predictions (Beaver 1966; Altman 1968; Altman et al. 1977; Platt and Platt 1990). Because of their relatively high discriminatory power, they are well accepted by the industry, even if they present some disadvantages, such as the lack of a theory that may explain why and how certain financial factors are linked to corporate bankruptcy.

While substituting the rigid SA with a more adaptive IRB increases the efficiency of credit allowance allocation, we are still facing a "standardized" use of IRBs. This paper contributes to increasing the flexibility of IRBs by introducing a methodology that focuses more on the specific components of the investment risk.

Given this unresolved puzzle, we also note that the literature does not investigate nor consider whether the merit of credit is correctly analysed and priced within the banks' financing system (Arzu et al. 2021). In fact, without answering this question, it is not possible to conclude whether the Basel regulations are the only culprit causing an inefficient

credit allocation between large and small companies. Indeed, severe measurement issues may arise regarding this topic, as Gleißner et al. (2022) clarify. The authors focus on the difficulties that may arise when exploring the long-term sustainability of financial performance, and then propose an innovative measure based on four conditions. While their approach can be inspiring thanks to the focus on long-term performance, our proposal is more concerned with the measurement of corporate risks and its nature.

We also note that little has been written on the issue of modelling credit risk specifically for private companies, chiefly SMEs. Moreover, the status quo on credit modelling for the banking system refers to Basel II rules, which use methods based on the concepts of probability of default, exposure at default, and loss-given default, which are well documented in the literature, starting with the seminal work by Beaver (1966), Altman (1968), and Ohlson (1980), and moving on to more recent work by Altman and Sabato (2007). Ciampi et al. (2021) provide a large literature review on SMEs' default prediction for the design of future perspectives. The research agenda they propose includes innovative approaches to exploiting new data sources using modern analytical techniques, such as artificial intelligence, machine learning, and macro-data inputs, with the aim of providing enhanced predictive results. Although these models are also well accepted by the industry, we think they present a major shortcoming: a horizon confined to 12 months ahead. Our proposal bypasses this time limit.

Accuracy in detecting the risk profile of SMEs can also be an issue, as Andreeva and Altman (2021) demonstrate. The authors compare risk assessment procedures for entrepreneurs/small business borrowers to those for consumers, when the same information on previous credit history is used for both types. They adopt cross-sectional logistic regressions and machine-learning models to discover superior methodologies for SME lending. They conclude that flexible models are required to improve the quality of SME loans, as they are able to derive more information from the endogenous components of their business risk. Our paper contributes to such a required flexibility by proposing a methodology that may be adapted to the contingent characteristics of the debtors.

Biases in credit-lending allocation in widely adopted methodologies have been investigated from an empirical perspective. For example, Altman et al. (2020) study the information asymmetries of the Italian mini-bond market and the consequent issues that the rating agencies may deal with in their "standard" procedures. From a similar perspective, Manelli et al. (2022) examine the pitfalls in connecting the merit of credit with growing opportunities by studying a sample of Italian SMEs. A possible proposal for solving the above puzzle is outlined by Roy and Shaw (2021) who suggest a multicriteria credit scoring model for SMEs using a hybrid best–worst method (BWM) and the technique for order of preference by similarity to ideal solution (TOPSIS). Some country-specific investigations highlight the major limits in evaluating the merits of SMEs, such as Kitowski et al. (2022) in the case of Poland, Kramoliš and Dobeš (2020) for the Czech Republic, Shinozaki (2015) for Asia and Gallucci et al. (2022) for Italy.

A commonality of these research articles is that private/SME lending is biased because successful ideas with a high profitability potential usually require a longer time horizon to develop. This can also bias the risk perception and its measurement using standard tools (e.g., the Von Neumann–Morgenstern approach), thus requiring upgrades in methodologies (Francis 2021). Therefore, we aim to contribute to the literature by proposing an innovative approach to assessing corporate values in incomplete markets (for private companies) through the identification of minimum required thresholds for corporate returns (e.g., to infer the pricing of bank loans). Provided that the most diffused practices are based on standard corporate finance frameworks for risk and the cost of capital assessments (for public companies), our proposal is based on an original extension of the certainty equivalent concept, as introduced by Lintner (1965). Hence, this study focuses on presenting a new methodology for rating and scoring companies, which finds its inner core in evaluating the asset-side capability of private SMEs to perform in the long run.

### 3. IRM: Our Alternative Model for Assessing the Return-to-Risk Performance of Firms

In applied corporate finance, the capital asset pricing model (CAPM) basics are still widely used to assess the required rate of return and the value of both equity and debt capital. All CAPM-based models suppose that investment assessment must be based on the marginal contribution of the asset to an already well-diversified portfolio, while the nature of risk is given (i.e., exogenous). Therefore, these models take into account (i) the asset sensitivity to non-diversifiable risk (i.e., systematic or market risk, often measured through the investment's beta), (ii) the expected return of the financial market and (iii) the risk-free rate. However, for private/unlisted companies, it is hard to estimate betas either through econometric computations or peer-group analyses. In fact, the required benchmarking process assumes the existence of a large quantity of comparable securities for sourcing the beta estimation (i.e., market efficiency) and the true tradability for any asset (i.e., market completeness). Standard scoring/rating systems are also based on this very unrealistic framework. These limits need to be bypassed to prevent significant underperformances in allocations, as demonstrated by the NPL levels.

Lintner's intuition (1965) is helpful in this regard, as he proposes assessing investments through certainty equivalent returns. Lintner demonstrates that the certainty equivalent approach can be used for investment valuation and comparison, since the resulting figures are fully consistent with those resulting from the Tobin's two-fund separation theorem and CAPM. However, his approach avoids complex estimations of risk premia, bypassing one of the most critical issues relating to standard methods.

Our proposal starts from the same intuitions of Gardenal (2010), who challenged Lintner's approach to SME evaluations through the measure of expected return and volatility. However, diverting from Gardenal's proposal, we prefer to apply Lintner's approach by following the original intuitions that underpin the "*competence value*" concept and the algorithms for computing the T-ratio (Mantovani 2014). This way, our proposal widens the use of Lintner's model by also considering the risk aversion of the single investor. This goal is achieved by using a shortfall approach in order to estimate a confident (instead of a certain) rate of return to score the long-term performance of the companies. The use of a shortfall approach allows for the inclusion of the investor's risk aversion through the confidence level while making IRM fully compliant with the Basel regulations. The scientific credibility of our proposal finds its roots in the zero-beta model (Black 1972). A description of the IRM proposal follows.

#### 3.1. Understanding the Methodological Limits to Bypass Standard Practices

In a standard neo-classic context, investors maximize their utility, given their degree of risk aversion, by choosing diversified portfolios on the capital market line (CML), as in Figure 2. The expected return (i.e., discounting rate) of each specific investment relates to the expected return of the chosen portfolio as a benchmark, while the market portfolio contains the investment under assessment.

Models using the above framework suppose that markets are efficient and, above all, complete, i.e., they are always capable of expressing a market price for any asset, including those under analysis. This represents a loophole, since the investment assessment methodology assumes that the investment has already been assessed. This is why best practices solve these flaws by assessing investment values and risk aversions related only to systematic risk, and then by assuming that any investment has a benchmarking process (peer groups) inside the market itself. The risk of a biased value assessment as a consequence of the methodologies one adopts for selecting the peer group can be unexpectedly high.

The key scientific advantage in Lintner's approach is the removal of the condition of complete markets for assessing the investment value. This is made possible by avoiding the estimation of the market risk premium. In fact: (i) the certainty equivalents of the expected cash flows are discounted (at the risk-free rate, while in CAPM, volatile cash flows are discounted at a risky rate); (ii) Lintner's approach is a total-risk measure estimation of the

CE, instead of a systematic (only) risk measure, as in the case of the CML portfolio models; (iii) finally, the methodology relies on accounting values, since it focuses on cash flows only, and then minimizes the use of market data.

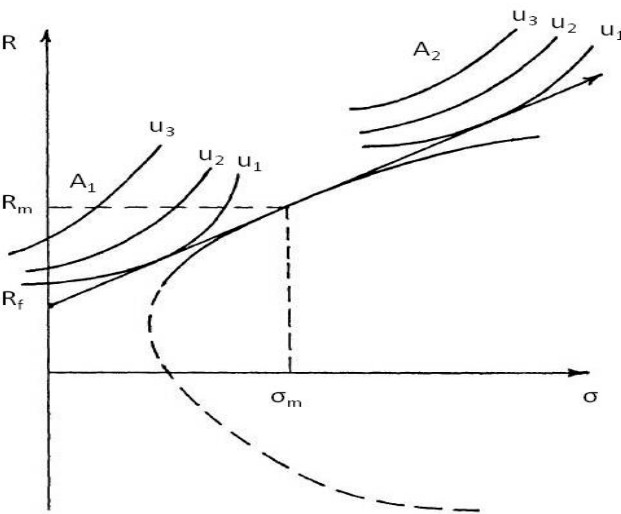

**Figure 2.** CML and investment choices depending on different risk aversion.

By comparing Lintner's approach with the CAPM for single investment evaluation, new insights may be found when assessing the value of unlisted companies. In a CAPM-based model, the relevant risk for the $i$-th asset is limited to only the systematic component of its variance, as described in Equation (1):

$$\sigma_i^2 = \beta_i^2 \sigma_m^2 + \sigma_\varepsilon^2 = \sigma_p^2 + \sigma_\varepsilon^2 \qquad (1)$$

where $\beta_i^2 \sigma_m^2 = \sigma_p^2$ is the systematic risk as expressed by the variance of a portfolio lying on the CML and with the same expected return of the $i$-th investment. $\sigma_\varepsilon^2$ is the firm-specific risk, which is supposed to be independent of systematic risk (i.e., no further addendums are required in the equation).

In efficient and complete markets, Tobin's two-fund separation theorem allows for a unique market portfolio to assess the risk premium of any investment, while the investor's risk aversion contributes only to selecting the portfolios on the CML. Given a generic investor's risk aversion ($A$), and a classic quadratic utility function (Von Neumann and Morgenstern 1953), the utility of the $i$-th investment is shown in Equation (2):

$$U_i = R_i - A\sigma_i^2 \qquad (2)$$

where $U_i$ stands for the achieved utility; $R_i$ is the expected return rate; $\sigma_i^2$ its volatility (i.e., variance); $A$ is a measure of the investor's marginal risk aversion.

In Lintner's framework, the same utility is achieved through a hypothetical risk-free investment prospecting an equivalent (i.e., "for-me-it's-the-same-to-have") risk-free return, ($R_F^*$), computed according to the investor's risk aversion. The "certainty equivalent" is shown in Equation (3).

$$U_i = R_F^* - A * 0 = R_F^* \qquad (3)$$

By substituting Equations (2) and (3), the certainty equivalent (i.e., $R_F^*$) then equals the difference between the expected return ($R_i$) and a portion of the volatility related to the investor's specific risk aversion ($A\sigma_i^2$). If $A > 0$ (i.e., the investor is always risk averse, as in CAPM and all the other models based on second-order stochastic dominance), then $R_F^*$ will always be higher than the risk-free return rate ($R_F$); in fact, such a gap (i.e., $R_F^* - R_F$) incentivizes the investor to switch to risky assets, according to her/his specific degree of risk tolerance. The higher the risk aversion, the wider the gap must be, while the

larger the gap between the certainty equivalent and $R_F$, the greater the asset value is for a given risk aversion.

Based on Equation (1), the investor's risk aversion (*A*) for the *i*-th investment is to be envisaged as consisting of two parts: the former relates to the systematic risk ($A_s$) and the latter to the asset-specific risk ($A_\varepsilon$). Accordingly, the investor's utility/certainty equivalent can be split, as in Equation (4):

$$U_i = R_i - A\sigma_i^2 = R_i - A_s\beta_i^2\sigma_m^2 - A_\varepsilon\sigma_\varepsilon^2 \tag{4}$$

The overall investor's risk aversion (*A*) is the weighted average of $A_s$ and $A_\varepsilon$, when the stochastic independence of the two sources of risk (systematic and firm specific) is assumed. However, no theories provide definitive conclusions about the relations between $A$, $A_s$, and $A_\varepsilon$ (Mantovani 1998). In fact, if the investor can really diversify their portfolio, the diversifiable risk becomes irrelevant, along with $A_\varepsilon$. An alternative explanation can also be considered: in large markets with massive volumes of transactions, the $A_\varepsilon$ of the different agents may clear each other. In other words, each transaction contributes (to a different extent) to a zero-utility game for the market as a whole.

*3.2. The Intuition: Substitute the Certainty Equivalent with the Confident Equivalent*

Our proposal is to apply a shortfall approach (Leibowitz and Henriksson 1989) to the framework originally proposed by Lintner (1965), to estimate the investment utility (i.e., its value) for a specific investor. In fact, the slope of the CML (i.e., the Sharpe ratio) relates to the investor's systematic-risk aversion, given a risk-free rate, when markets are complete. When markets are incomplete, $A_\varepsilon$ diverts from zero and the total risk aversion (*A*) must be considered. The risk-free rate is no longer the cornerstone that can be referred to in order to extract the utility as provided by the certainty equivalent.

Accordingly, any practical solution must (i) adopt a different cornerstone to the risk-free rate, (ii) consider the firm-specific risk and (iii) find a solution to estimate the total risk aversion "**A**". To deal with (i), one could use a CAPM-compliant approach, such as the zero-beta model by Black (1972). This allows the impact of the firm-specific risk to be embedded as well, if a short-fall approach is used to identify the investor's risk aversion. To deal with (ii) and (iii), we propose considering a **confident** **equivalent** return (i.e., a minimum return threshold that must be achieved according to a certain confidence level) instead of finding the certainty equivalent. At a methodological level, this approach simplifies the estimations, since the confidence level can be exploited ex ante (as in the Basel agreement).

Equation (5) below explains the gap between the expected return $E(R_i)$ for the *i*-th investment and the confident equivalent return (*Rce*), given a *j*% confidence:

$$E(R_i) = Rce + Z\sigma_i \quad => \quad Rce = E(R_i) - Z\sigma_i \tag{5}$$

where $j = \int_{-\infty}^{-Z} f(x)dx$.

As an example, Equation (5) suggests that when *j*% = 10%, the investor's risk aversion makes acceptable investments with ex post returns below *Rce* only once every 10 cases during the entire holding period.

It is important to point out the differences between the approach proposed here and the more classic concepts of risk aversion. In fact, the latter focus mainly on the return to risk ratio (usually at the marginal level and referring to the systematic risk only), while our approach focuses more on the loss tolerance for the entire time horizon of the investment. Our proposal is therefore consistent with the approaches supposing that the downside risk erodes more utility than the one created by the upside risk. Moreover, our shortfall-based approach is more dynamic, since the risk tolerance refers to the entire time horizon of the investment. Finally, while all investments fulfilling Equation (5) have the same characteristics that make them compliant with the investor's risk aversion, one given investment can be compliant with several shortfall lines/risk aversions.

Overall, given the above framework, the investor's choices can be accomplished according to three approaches:

(a)  $E(R_i) \geq R_f + S(\beta_i \sigma_m)$ i.e., the standard CAPM one ($S$ = Sharpe ratio = $\frac{R_m - R_f}{\sigma_m}$).
(b)  $E(R_i) \geq R_f^* + A\sigma_i^2$ i.e., the standard Lintner approach.
(c)  $E(R_i) \geq R_{CE} + Z\sigma_i$ i.e., the shortfall/zero-beta compliant approach proposed here.

### 3.3. From the Theoretical Framework towards a Methodological Implementation

From a theoretical point of view, the three approaches are equivalent if the markets are working well, i.e., they are at least efficient (possibly complete). In terms of a more practical outlook, the one proposed in this paper seems to be the easiest to adopt: no comparison with the peer group is required as in the (a) case, and no estimation of the risk aversion (i.e., the substitution rate between risk and return) is needed, as in case (b). In fact, for the (c) case, the estimation of the confident equivalent of a specific investment, given an ex ante confidence level, is sufficient: the investment choice will be a direct consequence of its comparison with the $R_{CE}$ being computed for the overall market. In fact, the market clears the contribution of any (non-zero) $A_\varepsilon$ and makes it possible to apply the (c) case even for incomplete markets, as in the case of unlisted and private companies. This also means that any investor may use this approach given its specific Z grade (e.g., the one fixed in the Basel agreement for the case of banks). Finally, by utilizing (c), one does not need to employ peer-group benchmarking and may focus mainly on accounting data; in fact, the only condition is to infer about the (whole) market-shortfall (Mantovani 2014).

Equation (6) below suggests an asset-side application of IRM to firms. Persistent returns from operations $P(ROI_i)$ are considered, along with their standard deviation ($\sigma_{ROI,i}$), given the Z-number matching the investor's risk aversion.

$$ROIce_i = P(ROI_i) - Z_i * \sigma_{ROI,i} \qquad (6)$$

The computed $ROIce_i$ can be used to rank private investments (including unlisted firms). The larger the $ROIce_i$, the higher the rank: accordingly, a score can be obtained by comparing $ROIce_i$ with the market confident equivalent as in Equation (7):

$$ROIce_i > Rce_m \qquad (7)$$

If $ROIce_i < Rce_m$ the company is underperforming in the long run, and therefore it has no creditworthiness. If Equation (7) is true, the larger the positive spread $ROIce_i - Rce_m$, the higher the merit of credit.

Alternatively, a threshold *ROI* [$T(ROI)$] can be computed and compared with $P(ROI)$, to make Equation (7) more useful for practitioners. The estimation of $T(ROI)$ is based on the market $Rce_m$ and standard deviation of *ROI* for the *i*-th firm. This approach seems more similar to that of the cost of capital, as Equation (8) depicts.

$$T(ROI_i) = Rce_m + Z_i * \sigma_{ROI,i} \qquad (8)$$

If expected *P(ROI)* is larger than *T(ROI)*, the company is compliant with the investor's risk tolerance; therefore, it creates long-term value. In fact, the spread $P(ROI_i) - T(ROI_i)$ provides the same selection and ranking of investments as in Equation (7), thus rating the firms' performance in the long run. A simple manipulation of Equation (7) demonstrates the equivalence with Equation (9). Based on Equation (7), we know that:

$$ROIce_i > Rce_m$$

$$ROIce_i + Z_i * \sigma_{ROI,i} > Rce_m + Z_i * \sigma_{ROI,i}$$
$$P(ROI_i) > T(ROI_i) \qquad (9)$$

Finally, the shortfall approach built into the IRM makes this approach compliant with the Basel principles.

For the practical use of Equations (6)–(9), one must estimate $\sigma_{ROI,i}$, information that may be obtained from due diligence for one specific investment, but that is very hard to infer for a wide sample (e.g., the credit portfolio of a bank). This is a direct consequence of the higher endogeneity that characterizes corporate risk compared to the market risk (Mantovani et al. 2013). Such endogeneity makes an estimation of $\sigma_{ROI,i}$ possible at the corporate level only, by adopting specific professional practices based on an integrated investigation of corporate risks.

The assumptions behind the confident equivalent of IRM imply a measure of $\sigma_{ROI,i}$ based on the components of the overall firm's endogenous risk as embedded into $P(ROI)$. In fact, $\sigma_{ROI,i}$ cannot be estimated by considering time series data only, because the volatility of firms' performance is not solely dependent on historical returns. Indeed, it depends on all (i.e., present and forthcoming) strategic decisions adopted by managers, as well as the corporate governance and management quality. This explains why corporate risk is more mean-reverting than market risk (Mantovani et al. 2013). Accordingly, proxy measures of firm strategy, as well as the governance and managerial decisions of unlisted companies, can be used to infer about $\sigma_{ROI,i}$ when they are the true drivers of $P(ROI)$.

For this purpose, the relationship as stated in Equation (10) below applies where the proxy indicators are based on financial reports, thus making this approach much more implementable for private investments, without losing the scientific affordability:

$$P(ROI_i) = \beta_0 + \beta_j * X_{j,i} + \epsilon_i \tag{10}$$

where $P(ROI)$ is the permanent return on investments; $\beta_0$ is the constant component; $X_j$ is the vector of $j$ variables, each measuring specific components of the corporate risk; $\beta_j$ is the vector of single risk relations for $j$ variables; and $\epsilon_i$ is a random component. The time subscript is omitted throughout.

If the relation in Equation (9) is sound, one can infer $\sigma_{ROI,i}$ by using the variance–covariance matrix of risk factors for a sample of firms, as in Equation (11).

$$\sigma_{ROI_i} = \beta_j * S_j * \beta_j^T + \epsilon_i \tag{11}$$

where $S_j$ is the sample variance–covariance matrix $j*j$ of the independent variables and $\beta_j^T$ the transposed vector of risk relations.

The same computations are possible for a single firm only if analyses are run to confirm the sound relations between the different components of the overall corporate risk (i.e., to detect the corporate variance–covariance matrix relating to the adopted managerial decisions). In any case, even with enough long time series, the relations estimated by Equation (11) are worthless in the case of a single firm. In fact, a firm is an entity in continuous evolution; because of the continuous changes in strategies, economic context, technologies, competitors, etc., the return-to-risk relation changes over time too. In this case, the direct estimation of Equation (8) is suggested.

The estimations of the return-to-risk relations over a sample of similar firms through Equations (10) and (11) can be useful for assessing the value of a portfolio of financial loans or the benchmarking of a specific firm with the strategies adopted by its peer companies. Once affordable evidence is found for $\sigma_{ROI,i}$ dependence on such relations, the above equations can be rearranged, as in Equation (12):

$$T(ROI_i) = R_{ce,i} + Z\sigma_{ROI_i} \tag{12}$$

$T(ROI_i)$ in Equation (11) is the same adopted in the Equation (8). Accordingly, it can be implemented for single firm rating by discovering the threshold ROI as in Equation (13).

$$T(ROI_i) = \beta_0 + \beta_j * X_{j,i} + \epsilon_i \tag{13}$$

Given the firm's $T(ROI_i)$ from Equation (13), one can compare it with the firm's actual level of P(ROI) then rate the firms accordingly and then create two clusters: the over-performers, (i.e., $P(ROI_i) > T(ROI_i)$) and the under-performers [$P(ROI_i) < T(ROI_i)$]. Essentially, a $P(ROI_i)$ that is higher than $T(ROI_i)$ is a firm with a positive return vs. risk mix.

## 4. How to Assess Creditworthiness through the IRM: An Empirical Investigation

In this section, we will show how the IRM could be used to jointly manage the quantity and pricing of credit allowances. The sign and magnitude of the gap $P(ROI_i) - T(ROI_i)$ is compared with: (1) $DEB/OPRE$, i.e., the debt intensity (as in Equation (14)), a proxy measure of the actual capability of the firms to raise the required resources; and (2) $INTE/DEB$, i.e., the cost of debt capital in Equation (15), which serves as a proxy for the risk premia adopted by the banking system.

$$Intensity\ of\ debt = DEB/OPRE_t = \frac{\left[(NFP_t^* + NFP_{t-1}^*)/2\right]}{OPRE_t} \tag{14}$$

$$Price\ of\ financing = INT/DEB_t = \frac{INTE_t}{\left[(GFP_t^* + GFP_{t-1}^*)/2\right]} \tag{15}$$

where

NFP = Net financial position = total debts − cash and cash equivalents.
OPRE = Operating revenue.
GFP = Gross financial position = loans + long-term debt.

Through the above comparisons, we highlight the differences between the Basel IRB methodologies (which define both the actual quantity and pricing of debt allowances) and our IRM outputs. On the one hand, companies with an above-average intensity of debt (Equation (14)) and below-average price of financing (Equation (15)) should be seen as the most creditworthy according to Basel's standards. On the other, companies with positive $P(ROI_i) - T(ROI_i)$ are the most affordable according to our IRM.

Given the empirical evidence on NPL dynamics and the several tentative suggestions for improving Basel's IRB, as discussed above, we expect to find mismatches for both quantity and pricing. Four different cross-section matches can be found for each indicator, with eight clusters in total. Table 1 below gives a detailed explanation of the eight clusters.

**Table 1.** Cross-section matches between IRM results and actual bank allowance distribution.

| Section A: P(ROI) − T(ROI) and DEB/OPRE classification | | | |
|---|---|---|---|
| | | **P(ROI) − T(ROI)** | |
| | | **Positive** | **Negative** |
| $DEB/OPRE_t$ | Higher | I. Firms with positive rating that raise more financial resources than sample average | II. Firms with negative rating that raise more financial resources than sample average |
| | Lower | III. Firms with positive rating that raise less financial resources than sample average | IV. Firms with negative rating that raise less financial resources than sample average |
| Section B: P(ROI) − T(ROI) and INTE/DEB classification | | | |
| | | **P(ROI) − T(ROI)** | |
| | | **Positive** | **Negative** |
| $INT/DEB_t$ | Lower | I. Firms with positive rating that pay less for their raised financial resources | II. Firms with negative rating that pay less for their raised financial resources |
| | Higher | III. Firms with positive rating that pay more for their raised financial resources | IV. Firms with negative rating that pay more for their raised financial resources |

The actual distribution of firms among the (critical vs. non-critical) cross-sections provides insights about the banks' ability to detect the risk components in firms' performance.

Figure 3 depicts the overall eight clusters and their ranking. Clusters #1 and #2 represent the highest allocative capability, for both quantities and pricing, of bank allowances, and are compliant with the results of the IRM. On the opposite side, clusters #8 and #7 show the lowest efficiency, and are either fully (#8) or substantially (#7) contrary to IRM suggestions. Finally, the remaining clusters (from #3 to #6) are ranked by their increasingly lower inefficiency.

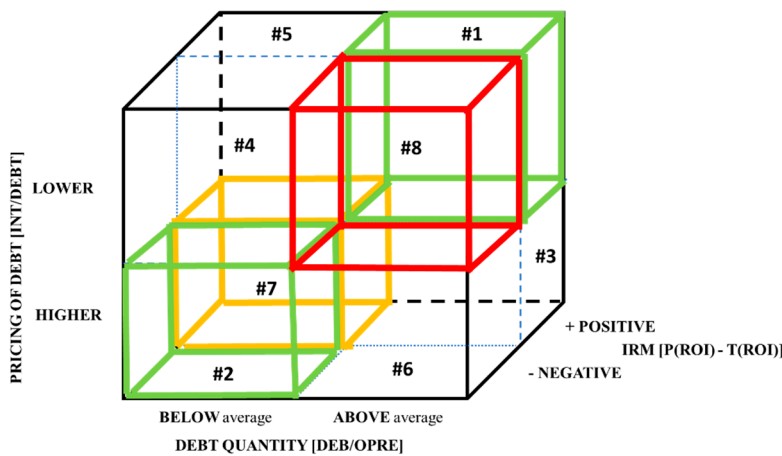

**Figure 3.** Overall cube-cross-section matches between IRM and bank allowances.

We back-test the IRM according to the schemes in Table 1 and Figure 3 for the period 2007–2014, which included both the Great Financial Crisis and the European sovereign debt crisis. To assess the IRM insights, the results of the back-test will be compared with the level of NPLs that was measured the year after the end of our sample period, and as reported by official statistics.

The empirical test is run using a sample of private manufacturing and services firms located in the north-east regions of Italy. While the Italian SMEs represent an international benchmark for their competitiveness and their distinctive hallmarks, the north-eastern firms suffer the most from inefficient bank financing. Moreover, such companies are mainly financed through debt capital provided by banks; accordingly, these inefficiencies may significantly impact the most performing forces behind economic growth in Italy.

The raw sample consists of 13,583 firms with headquarters in three regions of northeast Italy (Veneto, Friuli Venezia Giulia, and Trentino Alto-Adige), and are extracted from the Orbis database[4]. The sample selections are based on queries searching for firms with unconsolidated balance sheets for each year from 2007 to 2014. The first year is chosen according to the start of the financial crisis. For each company, we then consider a panel of eight years of financial data (i.e., 108,664 financial reports), to obtain at least seven useful figures for full IRM implementation (e.g., total assets, operating revenues, fixed assets, shareholders' funds, and costs of employees). Since some company profiles cannot provide the entire set of details as required for risk assessment according to the IRM, the final set is reduced to 12,431 firms. Table 2 provides the industry breakdown of our sample.

To compute $T(ROI)$ as in Equation (13), a panel regression is run on data from the eligible 12,431 firms for the entire time horizon, where the most significant indicators, i.e., those with the highest predictive power, are retained to estimate the confident equivalent of ROI. The dependent variable is represented by the return on investment ($ROI_{i,t}$), while the independent variables (the vectors $X_{i,t}$) are ratios typically used to profile the corporate risk. Autoregressive components are also considered.

$$ROI_{i,t} = \beta_0 + \beta_1 X_{i,t} + \beta_2 X_{i,t-1} + \varepsilon_{i,t} \tag{16}$$

where

*EBIT = Earnings before interest and taxes.*
*FIAS = Fixed assets.*
*WKCA = Working capital.*

$$ROI_t = \frac{EBIT_t}{(FIAS_t + WKCA_t + FIAS_{t-1} + WKCA_{t-1})/2}$$

In order to determine the optimal vector $X_{i,t}$ of risk factors, we consider three types of business risks and their operating, financial, and taxation indicators.

**Table 2.** Number of firms and key financial data by industry.

| Sectors-Definition | NACE Code Rev. 2 | Number | | Total Assets | | Operating Revenue | |
|---|---|---|---|---|---|---|---|
| | | Absolute | % | Absolute | % | Absolute | % |
| **MANUFACTURING** | | **7740** | **61.21%** | **519,996,368.85** | **67.20%** | **579,216,823.03** | **73.59%** |
| Manufacture of food products, beverages and tobacco products | 10; 11; 12 | 560 | 4.43% | 57,198,114.00 | 7.39% | 85,086,997.22 | 10.81% |
| Manufacture of textiles, apparel, leather and related products | 13; 14; 15 | 791 | 6.26% | 42,195,142.52 | 5.45% | 54,621,447.17 | 6.94% |
| Manufacture of wood and paper products and printing | 16; 17; 18 | 715 | 5.65% | 46,152,081.19 | 5.96% | 45,177,459.90 | 5.74% |
| Manufacture of coke and refined petroleum products | 19 | 13 | 0.10% | 781,687.40 | 0.10% | 1,016,381.95 | 0.13% |
| Manufacture of chemicals and chemical products | 20 | 207 | 1.64% | 16,280,064.67 | 2.10% | 19,285,752.88 | 2.45% |
| Manufacture of pharmaceuticals, medicinal chemical and botanical products | 21 | 14 | 0.11% | 3,299,509.57 | 0.43% | 3,400,539.27 | 0.43% |
| Manufacture of rubber and plastics products and other non-metallic mineral products | 22;23 | 887 | 7.02% | 61,207,649.54 | 7.91% | 57,556,877.22 | 7.31% |
| Manufacture of basic metals and fabricated metal products, except machinery and equipment | 24;25 | 1790 | 14.16% | 101,220,759.76 | 13.08% | 110,234,214.18 | 14.01% |
| Manufacture of computer, electronic and optical products | 26 | 200 | 1.58% | 10,323,986.84 | 1.33% | 11,307,819.33 | 1.44% |
| Manufacture of electrical equipment | 27 | 392 | 3.10% | 38,293,899.72 | 4.95% | 42,395,887.43 | 5.39% |
| Manufacture of machinery and equipment n.e.c. | 28 | 1066 | 8.43% | 78,689,218.42 | 10.17% | 82,224,883.47 | 10.45% |
| Manufacture of transport equipment | 29; 30 | 133 | 1.05% | 10,791,531.71 | 1.39% | 11,057,239.61 | 1.40% |
| Other manufacturing and repair and installation of machinery and equipment | 31; 32; 33 | 972 | 7.69% | 53,562,723.52 | 6.92% | 55,851,323.42 | 7.10% |
| **SERVICE** | | **4904** | **38.79%** | **253,799,100.59** | **32.80%** | **207,865,504.72** | **26.41%** |
| Agriculture, forestry and fishing | 01; 02; 03 | 341 | 2.70% | 28,219,668.48 | 3.65% | 28,939,922.01 | 3.68% |
| Mining and quarrying | 05; 06; 07; 08; 09 | 79 | 0.62% | 5,420,448.67 | 0.70% | 3,185,615.17 | 0.40% |
| Electricity, gas, steam and air-conditioning supply | 35 | 75 | 0.59% | 16,053,148.42 | 2.07% | 17,070,451.49 | 2.17% |
| Water supply, sewerage, waste management and remediation | 36; 37; 38; 39 | 202 | 1.60% | 16,663,958.39 | 2.15% | 13,183,644.09 | 1.68% |
| Transportation and storage | 49; 50; 51; 52; 53 | 936 | 7.40% | 46,420,204.99 | 6.00% | 50,042,648.69 | 6.36% |
| Accomodation and food service activities | 55; 56 | 676 | 5.35% | 30,467,807.26 | 3.94% | 14,774,800.39 | 1.88% |
| Publishing, audiovisual, broadcasting activities, telecommunications, IT and other information services | 58; 59; 60; 61; 62; 63 | 456 | 3.61% | 14,071,321.09 | 1.82% | 13,236,518.29 | 1.68% |
| Real estate activities | 68 | 362 | 2.86% | 31,027,385.17 | 4.01% | 11,586,698.44 | 1.47% |
| Legal, accounting, management, architecture, engineering, technical testing; analysis activities; scientific research and development; other professional, scientific and technical activities | 69; 70; 71; 72; 73; 74; 75 | 559 | 4.42% | 19,952,499.38 | 2.58% | 15,913,189.48 | 2.02% |
| Administrative support service activities | 77; 78; 79; 80; 81; 82 | 504 | 3.99% | 17,431,731.32 | 2.25% | 18,547,760.66 | 2.36% |
| Public administration and defence, compulsory social security | 84 | 0 | 0.00% | - | 0.00% | - | 0.00% |
| Education | 85 | 69 | 0.55% | 1,247,680.51 | 0.16% | 1,161,479.24 | 0.15% |
| Human health services, residential care and social work activities | 86; 87; 88 | 341 | 2.70% | 10,506,674.89 | 1.36% | 10,368,701.96 | 1.32% |
| Arts, entertainment and recreation | 90; 91; 92; 93 | 206 | 1.63% | 12,648,369.32 | 1.63% | 6,995,970.01 | 0.89% |
| Other services | 94; 95; 96 | 98 | 0.78% | 3,668,202.70 | 0.47% | 2,858,104.78 | 0.36% |

When dealing with private companies, the balance sheet is the sole instrument for inferring the firm's risk. We infer: (i) operating risk by applying risk indicators in relation to price (*DOL–price*), quantities (*DOL–volume*), technology (*CA/FIAS, WKCA/FIAS, RLFA, FIAS/OPRE*) and working capital (*WKCA/OPRE, CA/CL, CRED-DEBD*); (ii) financial risk through leverage (*DEB/OPRE, DEB/EQUITY, LEV*), debt maturity (*DEBLT*), ability to pay off debts (*DEB/EBITDA, EBIT/INT*), and interest rate (*INT/DEB*); (iii) combined financial and operating risks, by considering operative and characteristic cash flows (*FCFC/OPRE, FCFO/OPRE*); and (iv) that taxation risk is measured by tax to *EBIT* ratio (*TAX*). Additionally, we also consider some profitability indexes to capture the impact of managerial decisions as *AV/STAF* (a proxy for added value per employee) and *AV/EMPL* (as a proxy for added value per cost of employees), *ROS, ROE, ROI,* and *Adjusted ROI* (as a measure of ROI adjusted to stock variation)[5]. Table 3 below shows the final predictive regression for estimating *T(ROI)*.

**Table 3.** Predictive regression statistics.

| | Predictive Regression | | | | | |
|---|---|---|---|---|---|---|
| const | 0.0995 (0.0000) | *** | $DEB/EBITDA_{t-1}$ | −0.0002 (0.0649) | * |
| $CA/FIAS_{t-1}$ | 0.0002 (0.0000) | *** | $DEB/EQUITY_{t-1}$ | −0.0014 (0.0000) | *** |
| $CA/CL_t$ | 0.0131 (0.0000) | *** | $ROE_t$ | 0.0210 (0.0000) | *** |
| $FCFO/OPRE_t$ | 0.0108 (0.0000) | *** | $ROE_{t-1}$ | 0.0142 (0.0000) | *** |
| $DOL_t$ (volume) | −0.0001 (0.0509) | * | $EBIT/INT_t$ | 0.0000 (0.0000) | *** |
| $DOL_t$ (price) | −0.0008 (0.0022) | *** | $EBIT/INT_{t-1}$ | 0.0000 (0.0000) | *** |
| $DOI_{t-1}$ (price) | −0.0005 (0.0831) | * | $ROS_t$ | 0.4515 (0.0000) | *** |
| $FIAS/OPRE_t$ | −0.0172 (0.0000) | *** | $ROS_{t-1}$ | 0.0652 (0.0005) | *** |
| $FIAS/OPRE_{t-1}$ | −0.0130 (0.0000) | *** | $TAX_t$ | 0.0051 (0.0129) | *** |
| $INT/DEB_t$ | 0.0005 (0.0832) | * | $TAX_{t-1}$ | 0.0023 (0.0887) | * |
| $DEB/OPRE_t$ | 0.0180 (0.0002) | *** | $RLFA_t$ | −0.0001 (0.0605) | * |
| $DEB/OPRE_{t-1}$ | −0.0315 (0.0000) | *** | $RFLA_{t-1}$ | −0.0002 (0.0430) | ** |
| $DEB/EBITDA_t$ | −0.0002 (0.0389) | ** | | | |
| | | | R-squared | 0.0256 | |
| | | | Adjusted R-squared | 0.0256 | |
| | | | F-Stat (*p*-value) | 0.0000 | |

These are the standard symbols for *p*-value tests of the coefficient: *** *p*-value < 1%; ** *p*-value < 5%; * *p*-value < 10%.

The coefficients from Table 3 are used to calculate *T(ROI)* for each firm, which is then compared with the firm's actual level of *P(ROI)*, providing a ranking score and the eight clusters as explained above. Table 4 shows the resulting matrixes for the entire sample under analysis.

**Table 4.** Overlaps between IRM results, intensity of debt (left) and price of financing (right).

| | | IRM | | | Ranking | | | | | IRM | | |
|---|---|---|---|---|---|---|---|---|---|---|---|---|
| | | Positive | Negative | | # Firms | Clusters | % Firms | | | Positive | Negative | |
| **Intensity of debt** | higher | 3138 | 3689 | 6827 | 725 | 1 | 5.34% | **Price of financing** | lower | 4371 | 2391 | 6762 |
| | lower | 4947 | 1809 | 6756 | 479 | 2 | 3.53% | | higher | 3714 | 3107 | 6821 |
| | | 8085 | 5498 | **13,583** | 2413 | 3 | 17.76% | | | 8085 | 5498 | **13,583** |
| | | | | | 1330 | 4 | 9.79% | | | | | |
| | | IRM | | | 3646 | 5 | 26.84% | | | IRM | | |
| | | positive | negative | | 2628 | 6 | 19.35% | | | positive | negative | |
| **Intensity of debt** | higher | 23.10% | 27.16% | 50.26% | 1301 | 7 | 9.58% | **Price of financing** | lower | 32.18% | 17.60% | 49.78% |
| | lower | 36.42% | 13.32% | 49.74% | 1061 | 8 | 7.81% | | higher | 27.34% | 22.87% | 50.22% |
| | | 59.52% | 40.48% | **100.00%** | 13,583 | | 100.00% | | | 59.52% | 40.48% | **100.00%** |

**Note**. The red and orange colours in the ranking clusters can be summed to obtain similar coloured cross matches for quantity and pricing. In fact, by summing up cluster #8 with the left-hand, red-highlighted #6, one obtains the left-hand, red-coloured match, while summing it up with the right-hand, red-highlighted #4 provides the right-hand, red-coloured match. Similarly, one may sum cluster #7 with #5 to obtain the left figures or with #3 for those on the right. Green clusters (#1 and #2) represent those with the highest possible efficiency: indeed, they are subsets of the green areas on the left and right matches.

As per the quantity of bank allowances, we observe that 50.26% of Italian firms receive an above-average level of debt financing, while 59.52% of the firms have a positive IRM signal. However, only 23.10% of the firms are jointly creditworthy and receive adequate

bank allowances. Critical situations arise for the 27.16% firms which receive above-normal credit while having a negative IRM. Since this is a back-test, we may conclude that in 2014, these companies could be viewed as potential non-performing loans. Looking at actual data from 2015, one may observe that these figures are indeed compliant with the ones from official statistics, as illustrated in Figure 1.

In contrast, 36.42% of sampled firms should have received credit according to the IRM, when they did not in practice. Overall, misallocation applies to 63.58% of the firms in the sample (=27.16% + 36.42%).

Misallocations also emerge for the pricing of bank allowances. About 17.60% of firms underpaid their financial risks, while 27.34% had favourable ratings but overpaid for these resources, which then means that potential mispricing occurred almost half of the time. In conclusion, the above empirical evidence confirms that the Italian bank credit allocation is inefficient and therefore calls for significant improvement in credit rating and scoring.

Table 5 provides a breakdown of the results from Table 4, distinguished between the 7700 manufacturing and 4731 service companies included in the sample. According to Table 5, the IRM appears to be a selective method. In fact, by focusing on the second quadrant of the quantity matrix (i.e., the misallocation of credit leading to risk of default) we can see that manufacturing firms (32.08%) are worse off than services firms (19.62%). The very same conclusion applies for mispricing. According to the second quadrants of the two-cost matrix, services firms have lower frequencies (15.34%) than manufacturing ones (19.08%).

**Table 5.** Subset overlaps between IRM results, intensity of debt (left) and price of financing (right).

| Subset A: 7700 Manufacturing Firms | | | | | | | | | | | | |
|---|---|---|---|---|---|---|---|---|---|---|---|---|
| | | **IRM** | | | | **Ranking** | | | | **IRM** | | |
| | | Positive | Negative | | # Firms | Clusters | % Firms | | | Positive | Negative | |
| Intensity of debt | higher | 1491 | 2636 | 4127 | 381 | 1 | 4.64% | Price of financing | lower | 2529 | 1568 | 4097 |
| | lower | 2998 | 1092 | 4090 | 303 | 2 | 3.69% | | higher | 1960 | 2160 | 4120 |
| | | 4489 | 3728 | **8217** | 1110 | 3 | 13.51% | | | 4489 | 3728 | **8217** |
| | | | | | 789 | 4 | 9.60% | | | | | |
| | | **IRM** | | | 2148 | 5 | 26.14% | | | **IRM** | | |
| | | positive | negative | | 1857 | 6 | 22.60% | | | positive | negative | |
| Intensity of debt | higher | 18.15% | 32.08% | 50.23% | 850 | 7 | 10.34% | Price of financing | lower | 30.78% | 19.08% | 49.86% |
| | lower | 36.49% | 13.29% | 49.77% | 779 | 8 | 9.48% | | higher | 23.85% | 26.29% | 50.14% |
| | | 54.63% | 45.37% | 100.00% | 8217 | | 100.00% | | | 54.63% | 45.37% | 100.00% |
| **Subset B: 4731 Services Firms** | | | | | | | | | | | | |
| | | **IRM** | | | | **Ranking** | | | | **IRM** | | |
| | | Positive | Negative | | # Firms | Clusters | % Firms | | | Positive | Negative | |
| Intensity of debt | higher | 1647 | 1053 | 2700 | 344 | 1 | 6.41% | Price of financing | lower | 1842 | 823 | 2665 |
| | lower | 1949 | 717 | 2666 | 176 | 2 | 3.28% | | higher | 1754 | 947 | 2701 |
| | | 3596 | 1770 | **5366** | 1303 | 3 | 24.28% | | | 3596 | 1770 | **5366** |
| | | | | | 541 | 4 | 10.08% | | | | | |
| | | **IRM** | | | 1498 | 5 | 27.92% | | | **IRM** | | |
| | | positive | negative | | 771 | 6 | 14.37% | | | positive | negative | |
| Intensity of debt | higher | 30.69% | 19.62% | 50.32% | 451 | 7 | 8.40% | Price of financing | lower | 34.33% | 15.34% | 49.66% |
| | lower | 36.32% | 13.36% | 49.68% | 282 | 8 | 5.26% | | higher | 32.69% | 17.65% | 50.34% |
| | | 67.01% | 32.99% | 100.00% | 5366 | | 100.00% | | | 67.01% | 32.99% | 100.00% |

Note. Red and orange colours in ranking clusters can be summed to obtain similar coloured cross matches for quantity and pricing. In fact, by summing up cluster #8 with the left-hand, red-highlighted #6, one can obtain the left-hand, red-coloured match; while summing it up with the right-hand, red-highlighted #4, one can obtain the right-hand, red-coloured match. Similarly, one may sum cluster #7 with #5 to attain the left-hand figures or with #3 for those one on the right. Green clusters (#1 and #2) represent those with the highest possible efficiency: indeed, they are subsets of the green areas on the left and right matches.

Based on these data, it is difficult to conclude if the superior score for service firms is due to the superior allocative capability of the banking system or a consequence of simplified banking practices based on the preference to finance tangible assets (in manufacturing) instead of intangibles (in services). Nevertheless, we may confirm IRM selectivity by comparing the results from Table 5 with the figures of 10% $ROI_{ce}$, as computed for the subsample in Table 6.

**Table 6.** Confident equivalent breakdown.

| # of Firms | 12,431 | 7700 | 4731 |
|---|---|---|---|
| | **Total** sample | **Manufacturing** subset | **Service** subset |
| P(ROI) | 0.0682 | 0.0735 | 0.0588 |
| σ(ROI) | 0.1322 | 0.1898 | 0.1447 |
| *$ROI_{ce,i}$ (10%)* | *−0.1013* | *−0.1699* | *−0.1267* |

## 5. Conclusions

This paper proposes an IRM, i.e., a scoring and rating methodology based on an original development of the certainty equivalent model (Lintner 1965). The IRM focuses on the "confidence" to achieve sufficient long-term corporate returns to satisfy the investor's risk aversion. Our IRM differs from alternative proposals in terms of risk assessment, due to the inclusion of both exogenous and endogenous risk. We show how this approach is useful for private and unlisted companies such as SMEs, where a low number of comparable firms are available to estimate the risk premium through standard financial models. In fact, if risks are the determinants of endogenous uncertainty of future returns, a certainty equivalent methodology can be used more efficiently than standard financial models. This methodology does not need the historical volatility of the firm's returns, but rather the firm's risk impact on future returns volatility. The estimation of such an impact is made through a predictive regression to integrate Lintner's (1965) intuition and consideration of corporate risks.

We back-test IRM over a sample of 13,583 manufacturing and services firms in northeast Italy and compare our results with the allocation of bank loans (both quantity and pricing) originating from Basel-compliant rating and scoring methodologies. We estimate that only 9 out of 100 firms received credit by paying an appropriate (in line with their IRM rating) cost of debt capital. Moreover, 27 out of 100 firms received bank allowances which cannot be justified by their IRM indicators.

This latter evidence is a dramatic confirmation of the NPLs official data, which allows us to conclude that our IRM could contribute to increasing the efficiency of the loan market. Accordingly, we also argue that Basel regulations, as adopted by Italian banks, do not properly consider risks and return rates for private Italian SMEs.

However, even if the above results are promising, the model needs further development. First, an examination of other possible subsample classifications is required. Second, a more fine-tuned measure of firms' volatility is necessary for more general use. Finally, the analysis of the subsamples suggests the need for a deeper investigation into the causes behind a better credit allocation for services firms. This also implies that further improvements in the estimation of the standard deviations of ROI and corporate risks could further improve the IRM. These directions are part of our current research efforts towards creating an IRM 2.0.

**Author Contributions:** Conceptualization, G.M.M. and G.G.; methodology, G.M.M. and G.G.; software, G.M.M.; validation, G.G.; formal analysis, G.M.M. and G.G.; investigation, G.M.M. and G.G.; resources, G.M.M. and G.G.; data curation, G.M.M. and G.G.; writing—original draft preparation, G.M.M.; writing—review and editing, G.G.; visualization, G.M.M. and G.G.; supervision, G.M.M. and G.G.; project administration, G.M.M. and G.G.; funding acquisition, G.M.M. All authors have read and agreed to the published version of the manuscript.

**Funding:** This research received no external funding.

**Institutional Review Board Statement:** Not applicable.

**Informed Consent Statement:** Not applicable.

**Data Availability Statement:** Original source of corporate data ORBIS database.

**Conflicts of Interest:** The authors declare no conflict of interest.

## Appendix A

**Table A1.** Indexes' label, formula and definition.

| Index | Unit | Formula Derived from Orbis | Definition |
|---|---|---|---|
| *Technology features* | | | |
| $CA/FIAS_t$ | % | $CUAS_t/FIAS_t$ | Current rate of assets |
| $CA/CL_t$ | % | $CUAS_t/CULI_t$ | Current equilibrium |
| $WKCA/FIAS_t$ | % | $WKCA_t/FIAS_t$ | Relative intensity of working capital |
| $FIAS/OPRE_t$ | % | $\dfrac{[(FIAS_t + FIAS_{t-1})/2]}{OPRE_t}$ | Absolute intensity of fixed assets |
| $RLFA_t$ | – | $\dfrac{[(FIAS_t + FIAS_{t-1})/2]}{DEPR_t}$ | Residual life of fixed assets |
| *Financial strategy* | | | |
| $DEB/EBITDA_t$ | – | $\dfrac{[(NFP_t^* + NFP_{t-1}^*)/2]}{EBTA_t}$ | Years for debt re-financing |
| $DEBLT_t$ | % | $CUAS_t/NFP_t^*$ | Long-term debt rate |
| $DEB/EQUITY_t$ | – | $NFP_t^*/SHFD_t$ | Debt-to-equity ratio |
| $GDEB/EQUITY_t$ | | $GFP_t^{**}/SHFD_t$ | Gross debt-to-equity ratio |
| $DEB/OPRE_t$ | – | $\dfrac{[(NFP_t^* + NFP_{t-1}^*)/2]}{OPRE_t}$ | Intensity of debt |
| $LEV_t$ | – | $\dfrac{OPPL_t}{OPPL_t - INTE_t}$ | Financial leverage |
| $INTE/DEB_t$ | % | $\dfrac{INTE_t}{[(NFP_t^* + NFP_{t-1}^*)/2]}$ | Financial interest rate |
| *Operating risks* | | | |
| $WKCA/OPRE_t$ | % | $\dfrac{[(WKCA_t + WKCA_{t-1})/2]}{OPRE_t}$ | Absolute intensity of working capital |
| $DOL - volume_t$ | – | $AV_t/OPPL_t$ | Degree of operative leverage on volume changes |
| $DOL - price_t$ | – | $\left[\dfrac{MDCU_t^{***}}{(MDCU_t^{***} - x)} - 1\right] * 100$ | Degree of op. lev. on price changes of x (x = 1%) |
| $CRED - DEBT_t$ | dd | $\dfrac{(CRED_t + CRED_{t-1})/2}{MATE_t/365} - \dfrac{(DEBT_t + DEBT_{t-1})/2}{OPRE_t/365}$ | Difference between delays on payments to creditors and payments from debtors |

**Table A1.** *Cont.*

| Index | Unit | Formula Derived from Orbis | Definition |
|---|---|---|---|
| *Rate of return* | | | |
| $ROI_t$ | % | $\dfrac{OPPL_t}{\left[(CIN_t^{***} + CIN_{t-1}^{***})/2\right]}$ | Return on investment |
| Adjusted $ROI_t$ | % | $\dfrac{EBTA_t - STOK_t + STOK_{t-1}}{\left[(CIN_t^{****} + CIN_{t-1}^{****})/2\right]}$ | Alternative return on investment |
| $ROE_t$ | % | $\dfrac{PL_t}{\left[(SHFD_t + SHFD_{t-1})/2\right]}$ | Return on equity |
| $ROS_t$ | % | $OPPL_t/OPRE_t$ | Return on sales |
| $AV/STAF_t$ | % | $AV_t/STAF_t$ | Work productivity (cost of employees) |
| $AV/EMPL_t$ | % | $AV/EMPL_t$ | Work productivity (number of employees) |
| $EBIT/INT_t$ | – | $OPPL_t/INTE_t$ | Interest coverage |
| $FCFC/OPRE_t$ | % | $\dfrac{EBTA_t + WKCA_{t-1} - WKCA_t}{OPRE_t}$ | Margin of free cash flow characteristic |
| $FCFO/OPRE_t$ | % | $\dfrac{FCFC_t - (DEPR_t + FIAS_t - FIAS_{t-1})}{OPRE_t}$ | Margin of free cash flow operative |
| $TAX_t$ | % | $TAXA_t/OPPL_t$ | Tax rate |
| *Self-elaborated account values *,**,**** | | | |
| $NFP_t$ | € | $LOAN_t + LTDB_t - CASH_t$ | Net financial position |
| $GFP_t$ | € | $LOAN_t + LTDB_t$ | Gross financial position |
| $MDCU_t$ | % | $AV_t/OPRE_t$ | Contribution margin |
| $CIN_t$ | € | $FIAS_t + WKCA_t$ | Total net investments |

## Notes

[1]   According to Cannari et al. (2011), these companies produce one fourth of the private gross domestic product, while one fifth of the national population resides and one third of total national exports originate in the area.

[2]   (Basel Committee on Banking Supervision 2014), Bank for International Settlements 2014.

[3]   (Basel Committee on Banking Supervision 2010), Bank for International Settlements 2010.

[4]   Bureau van Dijk provides complete balance sheet data in the global standard format for global companies.

[5]   See the Appendix A, Table A1, for a complete view of the indicators.

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
