# Peer review of "How to Rate the Financial Performance of Private Companies? A Tailored Integrated Rating Methodology Applied to North-Eastern Italian Districts"

_jrfm, doi:10.3390/jrfm15110493_

Round 1
Reviewer 1 Report
My comments are given below. I am also including my suggestions to improve the manuscript.
The originality of the paper needs to be further clarified. The novelty shall be justified by highlighting that the manuscript contains sufficient contributions to the new body of knowledge. Thus, the literature survey should be extended to 2021-2, and the reference numbers of the newly pertinent journal papers need to be more clearly introduced, one by one. Moreover, you are requested to discuss how your approach differs from the cited references, then, reinforce your article’s innovation, and to revise the Introduction and Conclusion of the article. The relevant thesis should be revised in terms of language. There are some spelling errors. The number of references in the relevant article is quite low. should be supported by the studies carried out in this field in recent years.
Author Response
Dear reviewers,
First of all, thank you for your comments. We really appreciate them because give suggestions for the true improvement of the paper. Unfortunately, the Summer break and some health issues for one of the Author let us release the revised version today, only.
This release of the paper is prepared according to the followings:
- Literature review is now extended including relevant papers for our topic up to 2022
- Discussion of the literature review is focusing more on the distinctive elements of IRM (i.e. the confident equivalent as compared to “standard” risk premia discovery)
- Section 3 is amended to clarify more the innovative elements of the proposed model and to make easier reading the empirical evidence in the forthcoming section
- Conclusions have been amended accordingly along with some correlated items into the introduction. We also try to include the possible developments of the methodology
Last but not least, this release benefits of an additional proof-reading round which should help to highlight more the concepts we got aware to be not so clear, thanks to your comments.
Reviewer 2 Report
The article discusses a topic important from the point of view of financial analyzes (financial risk modeling, credit ratings), which may be of practical importance, which significantly increases the value of this article. Therefore, it can provide banks with practical guidance on the feasibility of implementing the new financial assessment method in small businesses. The present study is clear and comprehensive and constitute a significant contribution to the research on the use of new methods in financial analysis.
The value of the work is enhanced by the large research sample on which the study was carried out and the approach that the researchers used in developing an innovative rating method, i.e. referring to the long-term performance of firms, and therefore to the corporate returns' persistence and solvency.
The introduction provides sufficient background and includes relevant references.
In my opinion, the purpose of the study is concise and complete. However, it would be reasonable to assume the research hypothesis that the Basel rules on which Italian banks rely are underdeveloped because they do not properly consider risks and rates of return for Italian private SMEs.
The literature is correctly selected and used in the work. The literature review is current and comprehensive. The proposed research method used in the paper is precisely described and reference is made to the relevant scientific (methodological) research conducted in this area.
The content is briefly described and contextualized in relation to the presented theoretical background and selected empirical research on this topic. Drawings / tables / diagrams are appropriate. The statements and conclusions are consistent with the research results.
It should be emphasized that the authors note the weaknesses of the proposed method (which could be the main accusation in the work), especially in the context of its implications for all SMEs (including refining the measure of variability, testing on other sub-samples).
I rate the work highly.
Author Response

(The authors gave the same response as above.)

Reviewer 3 Report
decent and in-depth study in this field.
Author Response

(The authors gave the same response as above.)

Round 2
Reviewer 1 Report
It is a contribution to the relevant field of study and I believe that it will guide the future studies. The article is acceptable as it is.